# The Effect of Leaf Traits on the Excitation, Transmission, and Perception of Vibrational Mating Signals in the Tea Leafhopper *Empoasca onukii* Matsuda (Hemiptera: Cicadellidae)

**DOI:** 10.3390/plants14071147

**Published:** 2025-04-07

**Authors:** Yao Shan, Qiuyi Yao, Qisheng Jia, Jiping Lu, Xiaoming Cai, Zongmao Chen, Lei Bian

**Affiliations:** 1Key Laboratory of Biology, Genetics, and Breeding of Special Economic Animals and Plants, Ministry of Agriculture and Rural Affairs, Tea Research Institute, Chinese Academy of Agricultural Science, 9 Meiling South Road, Xihu District, Hangzhou 310008, China; shanyao0711@163.com (Y.S.); 13656696976@163.com (Q.Y.); qisheng.jia@outlook.com (Q.J.); violet_jiping@163.com (J.L.); cxm_d@mail.tricaas.com (X.C.); chenzm@tricaas.com (Z.C.); 2State Key Laboratory for Ecological Pest Control of Fujian and Taiwan Crops, Institute of Applied Ecology, Fujian Agriculture and Forestry University, Fuzhou 350002, China

**Keywords:** biotremology, leaf traits, plant-borne vibrations, *Empoasca onukii*, courtship communication

## Abstract

The physical properties of plants affect the transmission of plant-borne vibrational signals, which many herbivorous insects use for communication. Male calling signals (MCaSs, with sections S0, S1, and S2) and courtship signals (MCoSs, with sections S1 and S2) of *Empoasca onukii* Matsuda (Hemiptera: Cicadellidae), a major pest of tea plant, have a multicomponent structure. The same MCaS was repeatedly played back on different leaves of a tea branch, and parameters of the transmitted signal and female responses were measured on the leaf inhabited by females. We also measured the signal parameters and behaviors of *E. onukii* on single leaves of different ages. The intensity of MCaSs from other leaves attenuated after they propagated to leaves on which females were located, which decreased the duration of MCaS-S2. Higher leaf thickness, leaf hardness, and leaf area were associated with an increased pulse repetition time (*PRT*) of MCaSs, number of pulses in MCaS-S2, and duration of MCaS-S2, respectively. MCoS-S1 had a higher dominant frequency (*Df*) in leaves with a long main vein and high hardness, and the *PRT* of MCoS-S2 was longer on thicker leaves. In the initial stage of courtship, the signal excitation of males was affected by leaf traits, especially the temporal parameters of MCaS-S2, which was the most significantly affected section after host transmission; it also had an important effect on the response delay of females. In the location stage, the signal excitation of males was not only affected by leaf traits but also interacted with the signal excitation of females. These results facilitate exploration of the interaction between leafhoppers and host plants during courtship communication and have implications for the breeding of *E. onukii*-resistant varieties.

## 1. Introduction

Many insects use plant-borne vibrational signals in their courtship communication, which includes signal excitation, transmission, and perception [1,2]. Plants are highly complex structures and exhibit high variability in traits that can affect the transmission of vibrational signals [3], which has important consequences for the insects that inhabit them [4,5].

In recent years, the effect of variation of plant traits on the transmission of vibrational signals or cues has received increased attention [6]. The physical constraints of host plants affect vibrational signals, including the loss of energy, filtering, or distortion of information in signal transmission [7]. Passive variation in characteristic parameters of vibrational signals (intensity, frequency spectrum, and time pattern) potentially affects the detection and assessment of signals or cues by the receiver, such as decreasing mate attraction [8]. Some physical properties of substrates are inherently fixed, whereas other properties and parameters can be selected by animals. For example, organisms on plant stems can influence bending waves by selecting plants with particular dimensions, material properties, and generated vibration frequencies [7]. The behavioral adaptations of animals for generating vibrations can promote information transfer and thus mitigate physical constraints. For example, the treehopper *Enchenopa binotata* Say (Hemiptera: Membracidae) and the jumping spider *Habronattus dossenus* Griswold (Aranea: Salticidae) can alter their behaviors to tailor generated signals to substrate properties, which can be modulated by feedback during signal excitation [9,10]. An understanding of the physical mechanisms is important for addressing biological questions relating to communication and information gathering.

Our previous studies have shown that the tea leafhopper *Empoasca onukii*, a major monophagous sucking pest of tea plants, prefers to feed on tender stems and leaves [11]; however, leafhoppers prefer to send mating signals during courtship on mature leaves [12]. The standard shape and canopy structure of tea trees, a perennial and ligneous plant, have been affected by artificial cultivation [13]. As leaf age increases, some physical traits of the leaves on the branches of the canopy vary significantly, including leaf area, thickness, hardness, and color. Plant-borne vibrational signals are the key cues used in the courtship communication of *E. onukii* [12]. The courtship of *E. onukii* comprises five conserved stages: (1) call-fly: males broadcast specific calling signals (MCaSs), which serve as calling signals for potential females on tea plants; (2) female identification: females perceive MCaSs on the host and send signals (FS1) to respond to males; (3) male location: the male perceives FS1 and sends specific courtship signals (MCoSs), which form a closed-loop pattern with female signals (FS2s); (4) courtship: when the male is close enough to females, it sends continuous MCoSs, ignoring the FS2s; and (5) copulation: the male jumps up and copulates with the female.

To enhance our understanding of the interaction between *E. onukii* and host plants in courtship communication, we designed two experiments to quantify the variability in mating signals and behaviors caused by changes in leaf traits/age. We speculate that variability in leaf traits affects the excitation and transmission of mating signals of male *E. onukii* and indirectly affects female signal perception and behavior. Different sections of the male signals may carry different information and play special roles in mating communication. When an important section of the male signal is changed due to leaf traits, it may cause the female to respond differently.

## 2. Results

### 2.1. Female Responses to Signals from Other Leaves

#### 2.1.1. Variation in MCaS Parameters from Leaves of Different Ages

The signal parameters of the same MCaSs excited in leaves of different ages and transmitted to female leaves (*i* = 7) through stems (Figure 1A) varied significantly (*N* = 120; Figure 2), including the signal intensity (*I*) of MCaS-S0, MCaS-S1, and MCaS-S2 (*p* < 0.001; Appendix A; Figure 2A,B) and the duration (*D*) of MCaS-S2 (*F*_5,114_ = 5.09, *p* < 0.001). Greater transmitted distances were associated with greater intensity attenuation in each section of the MCaS. Each section of the MCaS decreased in duration due to intensity attenuation (Figure 2D). Changes observed in other signal parameters were not significant (Appendix A).

#### 2.1.2. Effect of MCaS Parameters on Female Behavior

A total of 65 females responded to the MCaS playback out of a total of 120 tests performed. A comparison of the MCaS parameters from tests with female responses and non-female responses revealed that none of the MCaS parameters affected female responses (*p* > 0.05; Appendix A), which indicated that whether the female responds to the MCaS was only related to the female’s physiological state and was independent of the intensity of the MCaS and the duration of MCaS-S2.

In the 65 tests with female responses, MCaS playback on each leaf stimulated female responses. After attenuation, the intensity of each section of the MCaS and *D*_MCaS-S0_ were used as independent variables, which were significantly correlated with *N*_FS1_ (all abbreviations are listed in Table 1; Figure 3). The multiple linear regression model of *N*_FS1_ was significant (*F*_4,60_ = 2.765, *p* = 0.035), and the effect of *I*_MCaS-S0_ was significant (*t* = 2.012, *p* = 0.049). In the multiple linear regression model (*F*_1,63_ = 4.976, *p* = 0.029), *D*_MCaS-S1_ had a significant effect on *Delay*_ID_ (*t* = 2.231, *p* = 0.029). Therefore, the intensity of MCaS-S0 might ensure the sustained attraction of the MCaS to the female; long MCaS-S1 durations resulted in long response delay of females to MCaSs (Figure 4A).

### 2.2. Female Responses to Male Signals from the Same Leaf

#### 2.2.1. Effect of Leaf Age

As leaf age (*i*) increased, the leaf thickness (*F*_8,171_ = 29.308, *p* < 0.001; Figure 5A), leaf width (*F*_8,171_ = 4.381, *p* < 0.001; Figure 5B), main vein length (*F*_8,171_ = 4.063, *p* < 0.001; Figure 5C), leaf margin perimeter (*F*_8,171_ = 3.833, *p* < 0.001; Figure 5D), and leaf area (*welch*_8,69.928_ = 13.33, *p* < 0.001; Figure 5E) gradually increased, and it tended to stabilize after the third or fourth leaf under the bud; however, the leaf hardness gradually increased (*F*_8,171_ = 23.088, *p* < 0.001; Figure 5F), peaked at the sixth leaf (*i =* 6), and then decreased after the ninth leaf.

Significant differences in seven MCaS parameters—including the *Df* and *N*_pulse_ of MCaS-S0 and -S1, *D*_MCaS-S1_, *PRT*_MCaS-S2_, *D*_FS1_, and female *Delay*_ID_—were observed for pairs of leafhoppers that established identification duets (*N* = 70) on single leaves of different ages (*p* < 0.05; Appendix A). Significant differences in five MCaS parameters, *Df*, *D*_MCoS-S1_, *N*_pulse_, *PRT*, and *D*_MCoS-S2_, and two behavioral indexes, female *Delay*_LO_ and male *Interval*_LO_, were observed for pairs of leafhoppers that established localization duets (*N* = 70) on leaves of different ages (*p* < 0.05; Appendix A). Leaf age had no significant effect on the whole time spent in courtship (*t*_courtship_; *welch*_9,25.76_ = 0.95, *p* = 0.5) and the number of localization cycles (*N*_cycle_; *welch*_9,25.17_ = 1.15, *p* = 0.37).

#### 2.2.2. Effect of Leaf Traits on the Identification Stage

The first MCaS parameters of males, the signal senders, should be affected only by leaf traits in the identification stage. There was a positive correlation between leaf thickness and *PRT*_MCaS-S2_ (*p* < 0.05, Spearman’s; Figure 6A), but the multiple linear regression model of *PRT*_MCaS-S2_ was not significant (*F*_1,68_ = 3.16, *p* = 0.078). In the multiple linear regression model of *PRT*_MCaS-S0_ (*F*_1,68_ = 5.13, *p* = 0.025), the effect of leaf thickness was significant (*t* = 2.265, *p* = 0.025). In the regression model of *D*_MCaS-S2_ (*F*_1,68_ = 5.59, *p* = 0.019), two leaf traits with multicollinearity [VIF (main vein and margin perimeter) > 5] were eliminated, and the effect of leaf area was significant (*t* = 2.36, *p* = 0.019). Leaf hardness (*t* = 2.624, *p* = 0.01) significantly affected the *N*_pulse_ of MCaS-S2 in the regression model (*F*_1,68_ = 6.88, *p* = 0.01). Eliminating leaf hardness (VIF > 5), *PRT*_MCaS-S1_ was significantly affected by leaf thickness (*t* = 1.987, *p* = 0.049) in the regression model (*F*_1,68_ = 3.95, *p* = 0.049). Therefore, high leaf thickness resulted in a long *PRT* of each section of the MCaS; high leaf hardness resulted in more pulses in the MCaS-S2, and large leaf area resulted in long MCaS-S2 durations (Figure 4A).

Signal parameters of females, the signal receivers, should be affected not only by leaf traits, but also by MCaS parameters in the identification stage. *Df*_FS1_ was significantly affected by *Df*_MCaS-S0_ (*t* = 4.32, *p* < 0.001) in the multiple linear regression model (*F*_2,67_ = 11.4, *p* < 0.001). *D*_FS1_ was significantly positively correlated with leaf width and four MCaS parameters (*p* < 0.05; Figure 6), but the multiple linear regression model of *D*_FS1_ was not significant (*F*_5,64_ = 1.81, *p* = 0.12). Leaf thickness (*t* = 3.26, *p* = 0.001), *N*_pulse_ of MCaS-S1 (*t* = 2.21, *p* = 0.029), and *PRT*_MCaS-S1_ (*t* = 2.4, *p* = 0.018) significantly affected female *Delay*_ID_ in the multiple linear regression model (*F*_7,62_ = 5.61, *p* < 0.001). Therefore, the dominant frequencies of male and female signals were affected by each other. A higher leaf thickness, longer *PRT*, and higher *N*_pulse_ of MCaS-S1 led to increases in the response delay of females to the MCaS (Figure 4A).

#### 2.2.3. Effect of Leaf Traits on Localization Stage

In the localization stage, MCoSs and FS2s alternated in a closed-loop mode. The second MCoS was affected not only by leaf traits but also by FS2 parameters. The multiple linear regression model (*F*_5,64_ = 4.62, *p* = 0.001) showed that *Df*_MCoS-S1_ was significantly affected by the main vein length (*t* = −2.28, *p* = 0.024), leaf hardness (*t* = 2.48, *p* = 0.014), and *Df*_FS2_ (*t* = 2.5, *p* = 0.013). When *D*_FS2_ was used as the independent variable (Figure 7A), it had a significant effect on the *Df* (*t* = −2.37, *p* = 0.019), *N*_pulse_ (*t* = −4.49, *p* < 0.001), and duration (*t* = −3.5, *p* = 0.001) of MCoS-S2 in the linear regression model (*Df*, *F*_1,68_ = 5.6, *p* = 0.019; *N*_pulse_, *F*_1,68_ = 20.19, *p* < 0.001; *duration*, *F*_1,68_ = 12.24, *p* = 0.001). *PRT*_MCoS-S2_ was significantly affected by leaf thickness (*t* = 2.04, *p* = 0.043) and *D*_FS2_ (*t* = 1.98, *p* = 0.049) in the regression model (*F*_2,67_ = 3.46, *p* = 0.034). Similarly, FS2 parameters might be affected by leaf traits and MCoS parameters. *Df*_MCoS-S1_ (*t* = 2.401, *p* = 0.017) significantly affected *Df*_FS2_ in the regression model (*F*_4,65_ = 3.43, *p* = 0.01). *D*_FS2_ was significantly affected by the *N*_pulse_ of MCoS-S2 (*t* = −3.757, *p* < 0.001) in the multiple linear regression model (*F*_7,62_ = 5.07, *p* < 0.001). Longer main veins and harder leaves led to increases in the *Df* of MCoS-S1, and thicker leaves resulted in increases in *PRT*_MCoS-S2_. The signal parameters of females and males affected each other (Figure 4B).

*Delay*_LO_ indicated the response time of a female to an MCoS. When the leaf thickness, hardness, and *N*_pulse_ of MCoS-S2, *PRT*_MCoS-S2_, and *D*_MCoS-S2_ (VIF > 5) were used as the independent variables (Figure 7B), only the *N*_pulse_ of MCoS-S2 (*t* = −6.43, *p* < 0.001) had a significant effect on *Delay*_LO_ in the multiple linear regression model (*F*_4,65_ = 15.44, *p* < 0.001). *Interval*_LO_ indicated the feedback speed of males to FS2. When the *N*_pulse_ of MCoS-S1, *N*_pulse_ of MCoS-S2, *D*_MCoS-S1_, and *D*_MCoS-S2_ were used as the independent variables (Figure 7B), *D*_MCoS-S1_ (*t* = 5.32, *p* < 0.001) and *D*_MCoS-S2_ (*t* = 5.03, *p* < 0.001) had a significant effect on *Interval*_LO_ in the multiple linear regression model (*F*_4,65_ = 13.62, *p* < 0.001). When the *N*_pulse_ and *D*_MCoS-S1_ were used as independent variables, significant regression models were observed for *t*_courtship_ (*F*_2,67_ = 3.744, *p* = 0.029) or *N*_cycle_ (*F*_2,67_ = 4.25, *p* = 0.018), but the independent variables did not reach significance. Therefore, leaf traits had no effect on the time (*t*_courtship_) and efficiency (*N*_cycle_) required by males to locate females on a single leaf. More pulses in MCoS-S2 resulted in a longer response delay of females to MCoSs, and longer MCoS durations resulted in a slower feedback speed of males to FS2s (Figure 4B).

## 3. Discussion

Clarifying host–pest interactions during the courtship communication of *E. onukii*, an obligate pest of tea plants, has important implications for understanding the mechanisms of adaptation of *E. onukii* to its host. During the whole courtship process, parameters of mating signals of male or female *E. onukii* changed within a certain range (Appendix A). Some parameter variations may be affected by host filtering of signals, but the intraindividual variability in parameters indicated that individuals might actively alter the parameters of their emitted signals. Our findings revealed that tea leaf traits affected the signal excitation and transmission of *E. onukii*, which indirectly affected mating behaviors. In addition, the results indicated that each section of male signals may have different functions.

### 3.1. Effects of Leaf Traits on Mating Behavior of Leafhoppers

Under natural conditions, female and male *E. onukii* are not located on the same leaves in most cases, and mating signals need to be transmitted through the host plant (male leaf–stem–female leaf). According to the review of Mortimer (2017) [7], the filtering of frequency is the first physical constraint acting on information propagation. As the low-pass filter of vibrational signals, plant tissue is not conducive to the transmission of high-frequency (>5 kHz) signals [14]. The frequency range of *E. onukii* mating signals is less than 2 kHz, and the pulse in MCaS-S0 has the lowest frequency (<0.4 kHz) [12,15], which is conducive to transmission via the plant. After the mating signal propagates from one leaf to another, the intensity inevitably decreases as the transmission distance increases. The wave speed distorts the temporal pattern of the propagating vibrational signal because the wave speed is not always the same for all frequency components of a signal [7]. MCaS playback on each leaf stimulated female responses, indicating that signal transmission on the same branch did not cause the distortion; that is, the whole branch was within the “active space” of the MCaS if the female was located on the seventh leaf below the tea bud [16]. The physical constraints of host plants appear to have no effect on the transmission of mating signals of *E. onukii* if the male and female are in the same branch, which indicates that *E. onukii* is highly adapted to its host.

Leaf thickness is positively correlated with the attenuation speed of vibrational signals [6]. Energy consumption has important implications for insects [17], and the ability to locate mates rapidly can prevent predation [18]. Our results showed that increases in leaf thickness (1) lead to a longer *PRT* of male signals, which is less conducive to saving energy, and (2) prolong the response delay of females to male signals, which impedes the ability of males to rapidly locate females. Therefore, thick leaves may negatively affect the courtship communication of *E. onukii*, which is consistent with the conclusion that increases in leaf thickness improve resistance to leafhopper infestations [19]. The hardness of plant tissues affects the excitation and propagation of vibrational signals [20]. In the identification stage, *E. onukii* males need MCaSs to be transmitted sufficiently far so that females can detect them. Our results showed that males might increase the number of pulses in MCaS-S2 on leaves with high hardness, which tended to be the sixth or seventh mature leaves below the bud. The increase in leaf hardness might lead to more information being carried by MCaSs.

The duration of the localization stage is approximately equivalent to the entire courtship time; in this stage, male *E. onukii* continuously move and exchange signals with females. During this stage, the leaf hardness and main vein length affected the frequency of MCoS-S1, which is consistent with the model of de Langre (2019) [20]. The signals of *E. onukii* might be tuned on the basis of the physical properties of the substrate, as has been reported in various insects [10,21,22]. A coordinated reciprocal exchange of information is common in male–female arthropods relying on substrate-borne vibrational signals [23]. The frequencies of MCoS-S1 and FS2 interact with each other, and the length of FS2 affects many parameters of MCoS-S1, indicating that the MCoS excitation of males is affected by female signals, as well as leaf traits.

### 3.2. Potential Information Carried in Different Sections of Male Signals

Similar to information transfer in other modalities, vibrational signals provide information on the location, identity, and even state of the sender [24,25]. Insect mating signals often have a multicomponent structure, and different sections of the signal may carry different types of information [26,27]. For example, the low-frequency pulses in MCaS-S0 may be used to maintain the female’s attention, which is consistent with the finding that males frequently send pulses similar to MCaS-S0 during the identification stage.

The distance from the signal playback point to the recording point affected the peak frequency measurements, with lower peak frequencies recorded at larger distances [4,6]; thus, the *Df* of each section in the MCaS was slightly decreased after transmission, but this did not affect MCaS recognition by females. *N*_pulse_, *PRT*, and *D*_MCaS-S1_ affect the signal response delay of females; we speculate that MCaS-S1 transmits information on species identity. In the localization stage, the *Df* of MCoS-S1 was positively correlated with that of FS2, perhaps because the *Df* of MCoS-S1 was affected by leaf traits, which indirectly affected the *Df* of female signals. Similarly to the leafhopper *Aphrodes makarovi* Zachvatkin (Hemiptera: Cicadellidae), the duration of the last section of the male localization signal was positively correlated with the duration of the female signal and the response delay of females, which indicates that vibrational duetting may entail the exchange of more complex information than just temporal coordination [23]. Few studies have examined the functions of different sections of vibrational signals with a multicomponent structure, and hypothesized functions need to be verified by playback experiments.

## 4. Materials and Methods

### 4.1. Insect Rearing

Adult tea leafhoppers were collected from the Tea Research Institute, Chinese Academy of Agricultural Sciences (TRI, CAAS), Hangzhou, China (30.18° N, 120.09° E) from September to November (peak period of *E. onukii* occurrence in autumn) in 2023 and 2024. Following the method described by Shan et al. (2023) [12], the leafhoppers were reared on tea branches (Longjing 43 variety) in an insectary with a stable environment (L10:D14, 25 ± 2 °C, 75 ± 5% RH). The adults used for testing were from the second and subsequent generations. Sexually mature seven-day-old virgin *E. onukii* adults were used in subsequent experiments. Each leafhopper was tested only once.

### 4.2. Plants and Leaf Trait Measurements

After pruning in the spring, the buds in the tea trees formed under an apical dominant growth pattern, and new branches had one bud and several leaves in summer and autumn. The tea branches (Longjing 43 variety) used for feeding and tests were collected at the same time in autumn from TRI, CAAS. The branches were immediately planted in sponges full of water and then reared in an insectary without pests. Branches with 10 leaves were used for tests of mating signal playback. From the top to the bottom of a tea branch, leaves below the bud were numbered according to their age (*i* = 1, 2 … 10, Figure 1A). Six leaf traits were measured in the laboratory (Table 1)—leaf area (cm^2^), leaf thickness (mm), leaf width (mm), main vein length (mm), leaf margin perimeter (mm), and leaf hardness (g)—following the handbook for standardized measurements of plant leaf traits. The leaf area, leaf width, main vein length, and leaf margin perimeter were first measured using an Intelligent and High Accuracy Leaf Area Tester (YMJ-CHA3, Zhejiang Top Instrument Co., Ltd., Hangzhou, China). Leaf hardness was measured using a Texture Analyzer (Model TA.XT plusC, Stable Micro Systems Ltd., Surrey, UK) equipped with a needle probe (P/2N-2 mm). The pre-test, test, and post-test speeds were 1, 2, and 10 mm/s, respectively, and the trigger force was 0.05 g. Measurements were taken from eight points distributed on both sides of the main vein of the leaves. Leaf thickness was finally measured using calipers on the same eight locations on the leaf.

### 4.3. Signal Recording and Playback

Vibrational signals were recorded using a previously described method [12]. These signals included the playback MCaS and mating signals, MCaSs, FS1s, MCoSs, and FS2s (Table 1), which were recorded using a laser vibrometer (VGO-200, Polytec, Waldbronn, Germany); these signals were digitized at a 48 kHz sample rate and 16-bit depth, transmitted via the generator module (LAN-XI 3160, Brüel and Kjær, Virum, Denmark), and saved using BK Connect software 2019 (Brüel and Kjær). To maximize the signal-to-noise ratio, the laser beam was focused on a reflective sticker (5 mm^2^) on the tea leaf. Leafhopper behavior was recorded via webcams (C1000e, Logitech, Lausanne, Switzerland), which enabled us to simultaneously observe behaviors associated with the emission of vibrational signals on the computers (Figure 1).

In Section 4.4.1, a female was repeatedly stimulated with a pre-recorded MCaS from an *E. onukii* male. The playback signal (repeated MCaSs) was sent to the generator module by BK Connect and converted into electrical signals, which were then transmitted through a power amplifier (type 2718, Brüel and Kjær) to a mini-shaker (Type 4810, Brüel and Kjær). A conical rod attached to the mini-shaker was in contact with the lower lamina of the target leaf to transmit MCaSs (Figure 1). To ensure that the parameters of the playback MCaS on the leaf in contact with the conical rod were consistent for each test, we recorded the playback with the laser vibrometer and fine-tuned the sound file before the test until the MCaS had the desired properties [28]. Based on the mean intensity of MCaSs recorded in Section 4.4.2, the signal intensity was adjusted to 8 ± 2 μm/s using BK Connect and the power amplifier before each test. During each test, the vibrations in the tea leaves were monitored in real time. The playback stimulation and the signals emitted by *E. onukii* were distinguished by differences in the signal spectrograms.

### 4.4. Experimental Setup

All experiments were conducted at dawn, between 5:00 h and 8:30 h, or dusk, between 17:00 h and 21:30 h, to ensure optimal mating activity of *E. onukii* [12] in an anechoic and sound-insulated chamber at TRI, CAAS with the same environment as the insectary.

#### 4.4.1. Female Responses to Signals from Other Leaves

Repeated units of a pre-recorded MCaS (*duration* = 1.491 s; *Df* = 340.8 Hz) were used to synthesize a 5 min playback signal at 2.5 s intervals by Adobe Audition 2021 (Adobe Systems Incorporated, San Jose, CA, USA). The synthesized signal was played back from leaves of different leaf ages and transmitted to female leaves (*i* = 7) (Figure 1A). We determined the parameters of transmitted MCaSs, as well as the responses of female *E. onukii* (Table 1). The key parameters of MCaSs affecting female responses were analyzed.

First, we inserted the tea branch into the flower mud and kept it vertical. The reflective stickers were attached to the middle of the main vein on the front lamina of each leaf on the branch. Next, the intensity of the playback signal on the target leaf that was in contact with the conical rod of the mini-shaker was pre-adjusted. The playback was paused. Next, a female leafhopper was guided to the seventh leaf under the bud, where females preferred to mate [12]. After a 5 min acclimation period, the synthesized signal was played back for 5 min; the vibrational signals on the leaf that the female inhabited were recorded (signals FS1). Finally, we removed the female and attached a reflective sticker to the location of the female, we then played back the MCaSs and recorded the signals (signals MCaS) again. To prevent the insects from escaping, the experiments were performed within a clear plexiglass cylinder (20 × 20 × 35 cm). In each test, we played back the synthesized signal on the second, fourth, sixth, seventh, eighth, and tenth leaves under the bud (*i* = 2, 4, 6, 7, 8, and 10, respectively; Figure 1). A new female was used for each playback test. The tests were repeated on a total of 20 tea branches.

A total of 120 signal samples were recorded, and each signal sample included a ‘signals FS1’ and a ‘signals MCaS’; the 120 trials were divided into two categories: female responses and female non-responses. We measured three signal parameters [*N* = 120; *Df*, *D*, and *I*] of the first MCaS in each ‘signals MCaS’. In the female response trials, we determined two female behavioral indexes in each ‘signals FS1’: *N*_FS1_ and *Delay*_ID_ (Table 1). *N*_FS1_ was the total number of FS1 in a signal sample, which indicated whether the transmitted MCaS had an enduring attraction for females. *Delay*_ID_ was the time interval from the end of the MCaS to the beginning of the subsequent FS1 and indicated the response delay of females to signals from males.

#### 4.4.2. Female Responses to Signals from the Same Leaf

In this test, we recorded the mating signals of a male–female pair on a single leaf of different leaf ages. The petiole of a tea leaf was placed into flower mud filled with water to prevent withering and keep it horizontal. The reflective sticker was attached to the middle of the main vein of the tea leaf. A pair of leafhoppers was guided to the leaf, and the vibrational signals on the leaf were recorded at the same time until the male and female copulated. To prevent the leafhoppers from escaping, the setup was contained within a plastic 6 cm diameter Petri dish (Figure 1B). The first to tenth single leaves under the bud (*i* from 1 to 10) were used. A new pair of leafhoppers was used for each test, and tests with the leaves of each age were repeated seven times. A total of 113 pairs were tested, and 70 pairs mated. Pairs that failed to copulate within 20 min were regarded as failed tests.

After the tests, we measured six leaf traits: leaf area, leaf thickness, leaf width, main vein length, leaf margin perimeter, and leaf hardness (Table 1). The tests generated 70 signal samples. In each signal sample, considering the intraindividual variability in mating signal parameters of *E. onukii* (Appendix A), we collected signal parameters in the first identification duet and the second location duet (*N* = 70; Table 1) to determine the effects of leaf traits on signal parameters and the effects of signal parameters on leafhopper behaviors. In addition to the six leaf traits and signal parameters, the whole time (*t*_courtship_) spent in the courtship process in each signal sample was recorded. We also measured the first *Delay*_ID_ in the identification stage, the second *Delay*_LO_ and *Interval*_LO_ in the location stage, and the number (*N*_cycle_) of localization cycles in each signal sample. *Interval*_LO_ was the interval between two MCoSs and indicated the feedback speed of males to FS2 in the localization stage (Table 1).

### 4.5. Terminology and Data Analyses

The terms and acronyms were designated on the basis of signal structure, behavioral context, and previous studies [12,29].

Intensity analysis of vibrational signals was performed using BK Connect software 2019 with the following Fast Fourier Transform (FFT) parameters: Hanning window, 75% overlap, 12.8 kHz frequency span, and 8 Hz frequency resolution. The mean vibration velocity at the dominant frequency component within the playback signal was regarded as the signal intensity. The temporal parameters of the signals recorded were analyzed using Adobe Audition 2021 with the following FFT parameters: Hanning window, window length of 8192 samples, and 75% overlap.

#### 4.5.1. Female Responses to Signals from Other Leaves

The test in Section 4.4.1 involved only the identification stage of *E. onukii*. Using leaf age (*i*) as the categorical variable, we determined the effect of leaf age on the signal parameters of transmitted MCaSs by analysis of variance. One-way ANOVA was used when the data met assumptions of normality and homoscedasticity; otherwise, Welch’s ANOVA was performed (*p* < 0.05; SPSS 25.0, IBM Corp., Armonk, NY, USA).

We used correlation analysis and multiple linear regression to identify potential factors that might contribute to explaining variance in signal parameters or mating behavior indexes. To determine which MCaS parameters contributed to female behaviors, we designated MCaS parameters that were significantly correlated with female behaviors (*N* = 120, Spearman’s correlation analysis, *p* < 0.05; SPSS 25.0) as independent variables; we then determined the MCaS parameters that significantly affected female behaviors using multiple linear regression (*p* < 0.05; SPSS 25.0). The independence between samples was then assessed by a Durbin–Watson test (*p* > 0.05). If the standardized residuals did not meet the assumptions of homoscedasticity and normality, the variables were ln-transformed to meet these requirements. The multicollinearity of the independent variable was determined by the variance inflation factor (VIF < 5), and independent variables with high linear relationships were eliminated; a secondary analysis was conducted to finally identify significant variables (*p* < 0.05).

In addition, MCaS parameters between the two categories (female response and non-response) were compared using an independent sample *t*-test (two-tailed, *p* < 0.05; SPSS 25.0) to determine the effects of these parameters on female behavior.

#### 4.5.2. Female Responses to Signals from the Same Leaf

The test in Section 4.4.2 involved the identification and localization stages of *E. onukii*. Using leaf age (*i*) as the categorical variable, we determined the effect of leaf age on leaf traits and mating signals by analysis of variance, which was performed as described in Section 4.5.1.

To determine which leaf traits potentially affect signal parameters and which signal parameters contribute to variance in behavioral indexes, the correlations between leaf traits, signal parameters, and four behavioral indexes were first determined (*N* = 70, Spearman’s, *p* < 0.05; SPSS 25.0); next, we used the leaf traits that were significantly correlated with signal parameters as independent variables to determine the leaf traits that significantly affected signal parameters using multiple linear regression (*p* < 0.05; SPSS 25.0); finally, signal parameters correlated with behavioral indexes were used as independent variables to determine the potential signal parameters that significantly affected the behavioral indexes in the multiple linear regression (*p* < 0.05).

## 5. Conclusions

Herbivorous insects need to adapt to hosts to optimize their reproduction [10]. The significant correlation between host leaf traits and mating signal parameters, as well as between female and male signal parameters, indicated that leaf traits affect the excitation and transmission of mating signals of male *E. onukii* and indirectly affect female signal excitation and behavior. Host plant adaptations of *E. onukii* for generating vibrational signals promote information transfer and mitigate physical constraints. As a commercially significant crop [30], tea exhibits remarkable diversity in the physical properties of its leaves across different varieties. These variations manifest in characteristics such as color, size, shape, thickness, and hardness [13,31]. Our findings show that some leaf traits are not conducive to the mating communication of *E. onukii* and have implications for the breeding of varieties with resistance to this serious pest.

## Figures and Tables

**Figure 1 plants-14-01147-f001:**
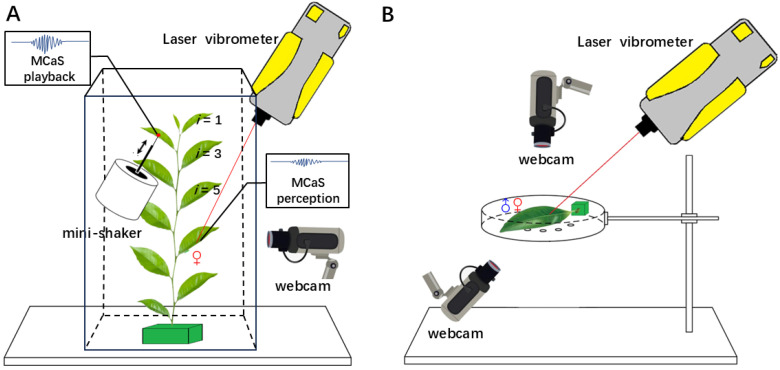
Experimental design. (**A**) The setup used for playback and monitoring male calling signals of *Empoasca onukii* from other leaves to the leaf inhabited by a female. The same male calling signals were played back via a mini-shaker from leaves of different ages (*i*, *i* = 1–10 from top to bottom) and propagated to the leaf that a female inhabited (*i* = 7). (**B**) The vibrational signals and behaviors of a pair of *E. onukii* adults on a single tea leaf were recorded simultaneously via a laser vibrometer and two webcams.

**Figure 2 plants-14-01147-f002:**
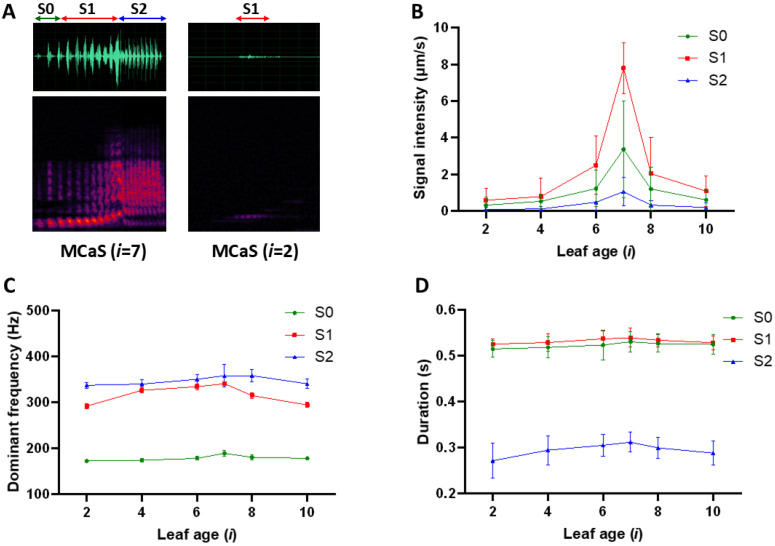
The variation in parameters of the same male calling signal (MCaS) after transmission from different leaves to the female on the seventh leaf below the tea bud. (**A**) Oscillogram (above) and spectrogram (below) of the same MCaS that was played back on the seventh leaf (left) or on the second leaf (right) below the tea bud and was recorded on the position inhabited by a female on the seventh leaf. *i* is the leaf age, which indicates the leaf stage of maturity (*i* = 1–10 from top to bottom). MCaS has a multicomponent structure and three sections: S0, S1, and S2. The intensity (**B**), dominant frequency (**C**), and duration (**D**) values (mean ± SD) of each section in the transmitted MCaSs at the location of the female on the seventh leaf below the tea bud.

**Figure 3 plants-14-01147-f003:**
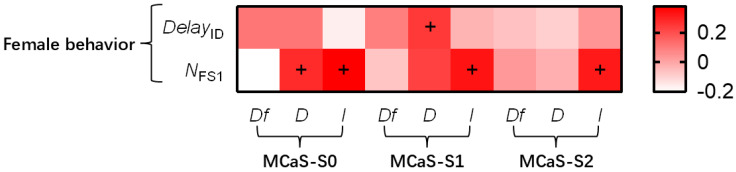
The correlation analysis between the parameters of transmitted MCaSs and female behavior. + indicates that an MCaS parameter was significantly positively correlated with female behavior (Spearman’s correlation analysis, *p* < 0.05). The abbreviations of signal parameters and female behavior indexes are shown in Table 1.

**Figure 4 plants-14-01147-f004:**
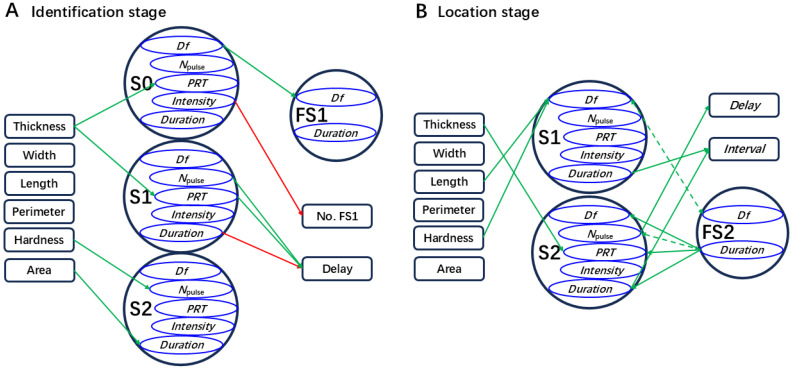
Diagram describing potential relationships between leaf traits and the signals and behavior of *Empoasca onukii*. In the identification stage (**A**) or the localization stage (**B**), a parameter of the male calling signal (MCaS) pointed to by the arrow indicates that this parameter is significantly affected by the leaf traits in the multiple linear regression model (*p* < 0.05); a parameter of the female signal or a behavioral index pointed to by the arrow indicates that this parameter or index is significantly affected by the MCaS parameters in the multiple linear regression model (*p* < 0.05). Red arrows are derived from the MCaS playback test in Section 4.4.1. Green arrows are derived from the signal acquisition test in Section 4.4.2. The dashed lines indicate that the factors significantly affect each other (*p* < 0.05). The abbreviations of signal parameters and female behavioral indexes are shown in Table 1.

**Figure 5 plants-14-01147-f005:**
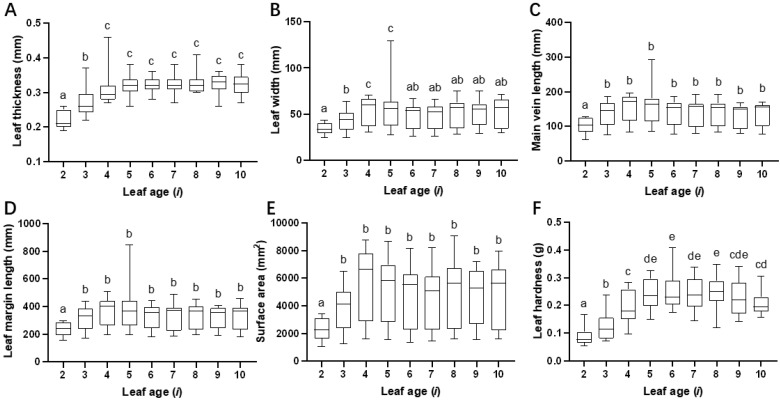
The effect of leaf age on six leaf traits. The effect of variation in leaf age (*i*) on leaf thickness (**A**), leaf width (**B**), main vein length (**C**), leaf margin perimeter (**D**), surface area (**E**), and leaf hardness (**F**). Each trait of leaves of different ages was compared by analysis of variance. One-way ANOVA and LSD tests were used when the data met assumptions of normality and homoscedasticity; otherwise, Welch’s ANOVA and Tamhane’s T2 (M) tests were performed (*p* < 0.05). Mean values with different letters indicate significant differences within the histogram.

**Figure 6 plants-14-01147-f006:**
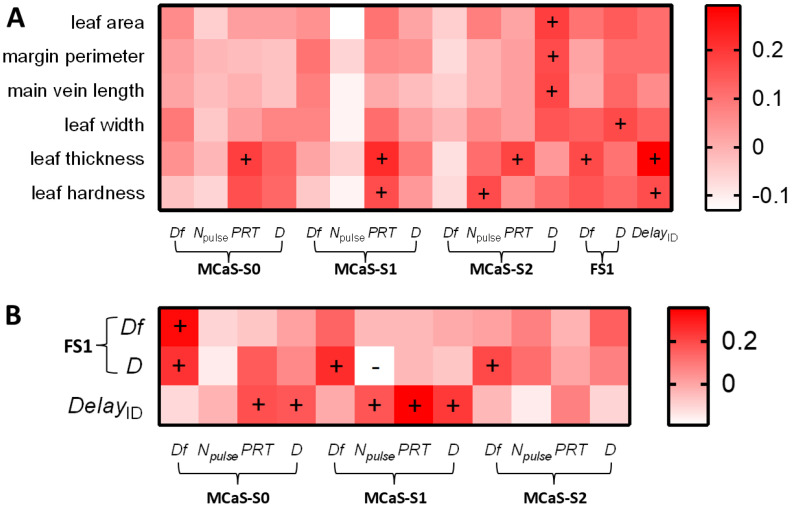
The correlation analysis between leaf traits, signal parameters, and female behavioral indexes in the identification stage of *Empoasca onukii*. (**A**) The correlations between six leaf traits and signal parameters or the female delay after the MCaS. (**B**) The correlation between the MCaS parameters and the FS1 parameters or the female delay after the MCaS. + and −, respectively, indicated that a factor was significantly positively and negatively correlated with another factor (Spearman’s correlation analysis, *p* < 0.05). The abbreviations of signal parameters and female behavioral indexes are shown in Table 1.

**Figure 7 plants-14-01147-f007:**
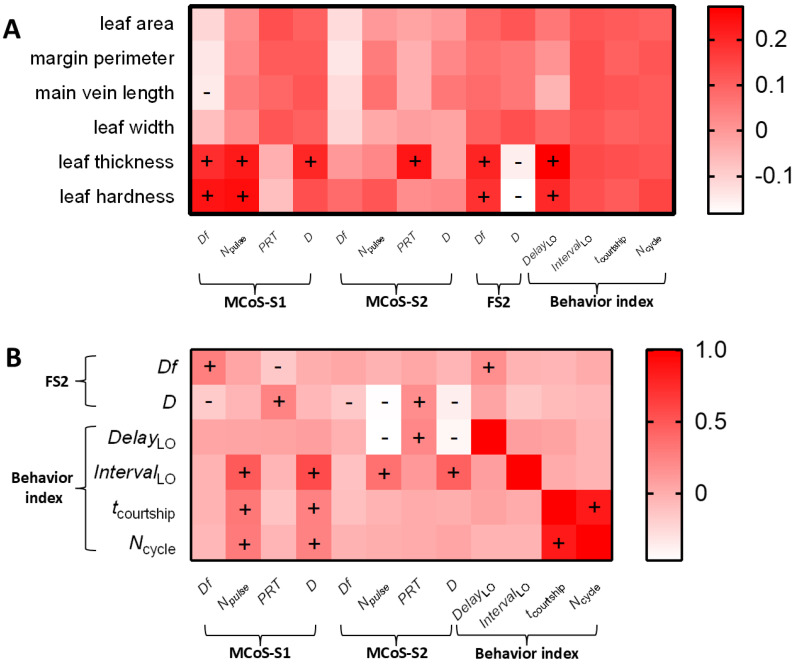
The correlation analysis between leaf traits, signal parameters, and behavioral indexes in the localization stage of *Empoasca onukii*. (**A**) The correlations of six leaf traits with signal parameters and behavioral indexes. (**B**) The correlations of the MCoS parameters with FS2 parameters and behavioral indexes. + and −, respectively, indicated that a factor was significantly positively and negatively correlated with another factor (Spearman’s correlation analysis, *p* < 0.05). The abbreviations of signal parameters and female behavioral indexes are shown in Table 1.

**Table 1 plants-14-01147-t001:** Abbreviations and data collected in the experiments.

Leaf Traits of Host Plants
Leaf area (cm^2^), leaf thickness (mm), leaf width (mm), main vein length (mm), leaf margin perimeter (mm), and leaf hardness (g). Leaf age (*i*), leaves from the top to the bottom of a tea branch were numbered.
Signal parameters
Male signal: male calling signal (MCaS, with three sections: S0, S1, and S2) and male courtship signal (MCoS, with two sections: S1, and S2). Female signal: female signal responses to the MCaS (FS1) or MCoS (FS2). Parameter: dominant frequency (*Df*), *PRT*, *N*_pulse_, duration (*D*), and intensity (*I*). *PRT*: pulse repetition time in each section of the MCaS or MCoS. *N*_pulse_: the number of pulses in each section of the MCaS or MCoS.
Behavioral indexes
Female behavior: *Delay*_ID_, *Delay*_LO_, and *N*_FS1_. Male behavior: *Interval*_LO_, *t*_courtship_, and *N*_cycle_. *Delay*_ID_: the time from the end of the MCaS to the beginning of the subsequent FS1. *Delay*_LO_: the time from the end of the MCoS to the beginning of the subsequent FS2. *N*_FS1_: the total number of FS1 in a signal sample in Section 4.4.1. *Interval*_LO_: the interval from the end of the MCoS to the beginning of the subsequent MCoS. *t*_courtship_: the whole time spent in courtship from the first MCaS to copulation in a signal sample in Section 4.4.2. *N*_cycle_: the number of localization cycles in a signal sample in Section 4.4.2. Some males needed to re-enter the identification stage several times during the localization process. We defined each time that the male entered the localization stage in the localization process as a localization cycle.

## Data Availability

Data are contained within the article and Appendix A.

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
