# Peer review of "The Effect of Leaf Traits on the Excitation, Transmission, and Perception of Vibrational Mating Signals in the Tea Leafhopper *Empoasca onukii* Matsuda (Hemiptera: Cicadellidae)"

_plants, 2025, doi:10.3390/plants14071147_

Round 1
Reviewer 1 Report
Comments and Suggestions for Authors
The study aims to test whether and how leaf traits modify the propagation of vibrational signals and whether the occurring changes affect the behavior of receiving individuals. Additionally, the authors aim to test whether insects can adjust the emission of vibrational signals based on the traits of leaves. Overall, the study is of interest for the biotremology community, which is growing worldwide, for basic and applied research. However, I think that there are at least two critical flaws in the methodology that must be addressed before publication.
1) The methodology section is unclear in many passages, but I could not find information on how the authors addressed the challenges of playback tests (well described along with the practical solutions in the chapter by Cocroft and colleagues “Vibrational playback experiments: challenges and solutions.” in Studying vibrational communication (2014): 249-274).
Also because of the well-known effects of substrates on vibrations’ propagation, it is critical in playback experiment to ensure the transmitted playback is consistent among replicates of the same test. In test 4.4.1, authors changed the position of the shaker multiple times, but they did not mention how they accounted for the filtering properties of the plant. One consequence of not compensating the playback is that the differences observed in the test could have been due to the different way the shaker was attached to the substrate.
2) Another part of the methodology that needs to be addressed is that authors always used different individuals in their tests. But by doing so, they cannot answer the question if the individuals are adapting their behavioral response to the leaf traits (for males) or to the received vibrations (for females). In fact, the observed differences can be due to individual variability. How did the authors account for this bias?
Below I list some minor issues that I think can be improved.
Abstract
In general, I think the abstract is missing a discussion and conclusion section that provide a wider perspective of the relevance of the results for the scientific community.
L24 What does PTR stand for? Is it the PRT defined at line 22?
Introduction
The first three references can be replaced with newer and more updated ones.
L38 All vibrations are subject to these modifications, so I suggest the authors to include here cues as well as signals.
LL43-45 This section is unclear; can you please rephrase it? Up to what extent do animals can adapt the vibrations they emit?
LL58-59: This statement seems bold. This is what we know up to now, but we cannot exclude that other communication modalities are involved. The history of biotremology should teach us at least to be cautious in describing the behavior of animals.
L61: I think that these cannot be defined “incidental cues” considering that they are emitted by males with the intent to inform females of their presence. They are signals, not cues. Cues and signals are specific terminology and must be used accordingly to their definition, they are not synonyms.
LL70-72: Is this one of the aims of the study? If so, I do not think that the methodology used is sufficient to answer this question. See my second main point at the beginning of my comment.
Figures
Figure1B: Why there is a shaker below the leaf?
Figure 2: From the figure it seems that the authors tested two different playbacks on different leaves, can you please clarify?
What are the values reported in the plots? Are they the mean and the standard error or the standard deviation?
Discussion
L225: What do we know about the possibility for males to adjust their signals parameters? Maybe you can add some details about the intraindividual variability of the emitted signals, otherwise it is very speculative to affirm that males can emit vibrational signals with different parameters.
Materials and methods
Throughout this section, authors use the term “samples” in an unclear way. I think they mean replicates or trials, I suggest changing the term to adjust it to the international consuetudine.
LL342-343: Since this is the first time that the amplitude of the playback is mentioned, the value used should be given here. Now, it is given at L360.
Additionally, the reason why a specific value of 8 ± 2 µm/s should be given.
LL360-362: Was the female present when the amplitude of the playback was adjusted? Please, clarify the order of the actions described here.
L363: From which leaf the vibrational signal was recorded?
L387: What are the six leaf traits?
L385-387: It is unclear if all the tested pairs mated or if you tested more than 70 pairs and only 70 did mate. Can you clarify this part, please?
LL397-406: What FFT parameters were applied for the analysis of spectral features of the signals?
Conclusion
The conclusion is very concise and narrow, and it lacks any reference to other insect-plant system. I think it can be improved by adding at least the relevance of the obtained results in the study of vibrational communication in plant dwelling insects and to what extent they can impact the management of insect pests.
Author Response
The study aims to test whether and how leaf traits modify the propagation of vibrational signals and whether the occurring changes affect the behavior of receiving individuals. Additionally, the authors aim to test whether insects can adjust the emission of vibrational signals based on the traits of leaves. Overall, the study is of interest for the biotremology community, which is growing worldwide, for basic and applied research. However, I think that there are at least two critical flaws in the methodology that must be addressed before publication.
Response: Thank you for your valuable and constructive suggestions on our research. These suggestions are important for improving the manuscript and will aid our future work. We carefully revised the manuscript according to your comments, including the usage of terms, descriptions of the methods, and conclusions.
We hope that the revisions and our responses have improved our manuscript.
- The methodology section is unclear in many passages, but I could not find information on how the authors addressed the challenges of playback tests (well described along with the practical solutions in the chapter by Cocroft and colleagues “Vibrational playback experiments: challenges and solutions.” in Studying vibrational communication (2014): 249-274.
Also because of the well-known effects of substrates on vibrations’ propagation, it is critical in playback experiment to ensure the transmitted playback is consistent among replicates of the same test. In test 4.4.1, authors changed the position of the shaker multiple times, but they did not mention how they accounted for the filtering properties of the plant. One consequence of not compensating the playback is that the differences observed in the test could have been due to the different way the shaker was attached to the substrate.
Response: We apologize for the confusion caused by our description of the method. In the field of biotremology, numerous studies on vibration signal playback, including our own research, have been based on the foundational work of Cocroft et al. (2014). We cited this chapter after revision. (LL 585-587)
In test 4.4.1, altering the position (leaves of different leaf ages) of the MCaS playback changed the parameters of the recorded MCaS on the seventh leaf inhabited by a female. We characterized the responses of females to the transmitted MCaS based on the changes in parameters. To ensure that the parameters of the playback MCaS at each position were consistent for each test, we recorded the playback with the laser vibrometer and fine-tuned the sound file before the test until the MCaS had the desired properties. This procedure has been supplemented in 4.3. (LL376-379)
- Another part of the methodology that needs to be addressed is that authors always used different individuals in their tests. But by doing so, they cannot answer the question if the individuals are adapting their behavioral response to the leaf traits (for males) or to the received vibrations (for females). In fact, the observed differences can be due to individual variability. How did the authors account for this bias?
Response: Thank you for your comments. Using different individuals does not provide direct evidence for the hypothesis that “leafhoppers adjust signal parameters according to the different microhabitats.” Thus, we removed the conclusive descriptions related to this hypothesis in the manuscript. (LL83-84; LL254-258; LL497-500) We hope the following statement can account for your bias.
We used different individuals for the following reasons.
[1] Employing different individuals as biological replicates in the experiment enhances the generalizability of the findings, ensuring that the conclusions are more universally applicable.
[2] On the basis of your suggestion, we confirmed the intraindividual and interindividual variability in mating signals recorded from the same leaf (Figure S1). Throughout the courtship process, the parameters of mating signals in both male and female E. onukii exhibit dynamic variations within a defined range. The use of different individuals as experimental replicates helps mitigate the impact of these variabilities, thereby enhancing the robustness of the findings.
[3] In 4.4.1, we used one female for all trials. However, owing to the memory of insects, repeated stimulation habituates the female to MCaSs, which causes the female to no longer respond to MCaS stimuli in subsequent trials. In 4.4.2, once the pair of leafhoppers mates, they cannot be separated, and the male and female lose their desire to mate for a long time after copulating.
In addition, variation in the parameters of signals emitted from an individual was not random. For example:
[1] The parameter Npulse of MCaS was highly variable. The Npulse of the first MCaS in the call-fly stage was always lower than that in the identification stage (male and female completed at least one identification duet), which is why we collected signal parameters of the MCaS in the first identification duet (not the first MCaS in the call-fly stage). Before the first identification duet, the male had already sent at least one MCaS and was not sure whether there were females nearby; so the parameters of this MCaS were affected by their own state and microhabitats.
[2] The male kept moving during the location stage, and the parameter DurationMCoS-S2 at the beginning of the location stage was always lower than that at the end of the location stage. The distance between males and females in the initial part of the location stage was greater than that at the end of the location stage; thus, DurationMCoS-S2 might be affected by the female signals.
[3] The position of females in the process of courtship usually did not change, and the intraindividual differences in their signal parameters were likely affected by male signals.
Below I list some minor issues that I think can be improved.
Abstract
- In general, I think the abstract is missing a discussion and conclusion section that provide a wider perspective of the relevance of the results for the scientific community.
Response: Thank you for your suggestion. We summarized the conclusion and significance of the study and added them to the Abstract. (LL26-34)
- L24 What does PTR stand for? Is it the PRT defined at line 22?
Response: We apologize for this oversight. This has been corrected throughout the manuscript.
Introduction
- The first three references can be replaced with newer and more updated ones.
Response: Thank you for your suggestion. The references have been updated.
- L38 All vibrations are subject to these modifications, so I suggest the authors to include here cues as well as signals.
Response: This has been revised as suggested. (L46)
- LL43-45 This section is unclear; can you please rephrase it? Up to what extent do animals can adapt the vibrations they emit?
Response: Thank you for this suggestion. This has been rephrased as ‘Some … frequency.’ (LL51-55)
- LL58-59: This statement seems bold. This is what we know up to now, but we cannot exclude that other communication modalities are involved. The history of biotremology should teach us at least to be cautious in describing the behavior of animals.
Response: This has been rephrased as ‘Plant-borne vibrational signals are the key cues used in the courtship communication of E. onukii’. (L68)
- L61: I think that these cannot be defined “incidental cues” considering that they are emitted by males with the intent to inform females of their presence. They are signals, not cues. Cues and signals are specific terminology and must be used accordingly to their definition, they are not synonyms.
Response: Thank you for this suggestion; ‘incidental cues’ has been replaced with ‘calling signals’. (L71)
- LL70-72: Is this one of the aims of the study? If so, I do not think that the methodology used is sufficient to answer this question. See my second main point at the beginning of my comment.
Response: This statement is not the aim of our study, but a conjecture. This statement has been removed in the revised version. (LL83-84)
Figures
- Figure1B: Why there is a shaker below the leaf?
Response: This error has been corrected.
- Figure 2: From the figure it seems that the authors tested two different playbacks on different leaves, can you please clarify? What are the values reported in the plots? Are they the mean and the standard error or the standard deviation?
Response: We apologize for any confusion associated with this. We played back the same MCaS on different leaves and recorded it at the location of the female on the seventh leaf below the tea bud, and the values reported in the plots were mean ± SD. We revised the legend to better clarify this information.
Discussion
- L225: What do we know about the possibility for males to adjust their signals parameters? Maybe you can add some details about the intraindividual variability of the emitted signals, otherwise it is very speculative to affirm that males can emit vibrational signals with different parameters.
Response: Thank you. On the basis of your suggestion, we analyzed the intraindividual variability in signal parameters emitted by individuals in the courtship process. The results are presented in Figure S1.
‘During the whole … their emitted signals.’ This statement has been added to the discussion. (LL259-262)
Materials and methods
- Throughout this section, authors use the term “samples” in an unclear way. I think they mean replicates or trials, I suggest changing the term to adjust it to the international consuetudine.
Response: Thank you. In most cases, we used ‘signal sample’ to represent the whole signal recorded from each test. Except for ‘signal sample’ and other specific terms, we replaced ‘sample’ with ‘trial’ according to your suggestion.
- 15. LL342-343: Since this is the first time that the amplitude of the playback is mentioned, the value used should be given here. Now, it is given at L360.
Response: This has been revised as you suggested. (L380)
- Additionally, the reason why a specific value of 8 ± 2 µm/s should be given.
Response: Thank you. This value was used based on the mean intensity of MCaSs recorded in 4.4.2. This was revised as suggested. (LL379-380)
- 17. LL360-362: Was the female present when the amplitude of the playback was adjusted? Please, clarify the order of the actions described here.
Response: The female was not present when the signal amplitude was adjusted. We have revised the details of the actions as suggested. (LL399-402)
- 18. L363: From which leaf the vibrational signal was recorded?
Response: Apologies for any confusion associated with this. The vibrational signals were recorded on the leaf that the female inhabited, which was supplemented after revision. (LL403-404)
- 19. L387: What are the six leaf traits?
Response: The six leaf traits were supplemented after revision. (LL431-432)
- 20. L385-387: It is unclear if all the tested pairs mated or if you tested more than 70 pairs and only 70 did mate. Can you clarify this part, please?
Response: We tested 113 pairs, and only 70 mated. This information was added. (LL429-430)
- 21. LL397-406: What FFT parameters were applied for the analysis of spectral features of the signals?
Response: This confusing statement has been revised. (LL447, 451)
Conclusion
- The conclusion is very concise and narrow, and it lacks any reference to other insect-plant system. I think it can be improved by adding at least the relevance of the obtained results in the study of vibrational communication in plant dwelling insects and to what extent they can impact the management of insect pests.
Response: Thank you for your suggestion. We deleted the redundant statements about the details of the results in the conclusion and added information on the relevance of the results and some implications for the management of this pest.

Reviewer 2 Report
Comments and Suggestions for Authors
This paper is innovative. I think this manuscript should be published in the journal after revision.
Abstract:
L14:Delete ‘(e.g., Empoasca onukii)’.
L15:Supplement the host after Empoasca onukii.
Introduction:
L42:I think this should be ‘time pattern’her.
Results
L77:Indicate the position of the leaf that female inhabited.
Figure 1. Behavioral monitoring is not involved in Figure 1A, and supplement the position of the leaf that female inhabited.
L102:‘and was independent of MCaS parameters.’ I think this sentence should be changed to 'intensity of MCaS and duration of MCaS-S2', because the frequency does not change significantly after transmission.
L105:I didn't find Table 1 in the manuscript. Why is Table 1 in the supplementary materials? This has a serious impact on the revision of the manuscript.
L108-110:Conjectural conclusions need to be formulated with care, after all, the function of these sections has not yet been verified.
L131:Here should be the second or third leaf under the bud.
L134:Replace ‘signal’ with ‘MCaS’.
L137:Replace ‘signal’ with ‘MCaS’.
L151-152:Are males always the ones initiating courtship communication?
L171:The case of the acronym 'delay' should be unified (capital and lower-case letter), as should similar issues with other acronyms.
L192:Replace ‘PTRMCoS-S2’ with ‘PRTMCoS-S2’.
L199-200:Simplify this sentence ‘The signal parameters… by female signals’.
L201:Delete ‘DelayLO indicated the response time of a female to an MCoS’.
Discussion
L223-224: I don't understand this sentence, the causal relationship seems to be wrong
L227:Replace ’signal excitation’ with ‘signal excitation of males’.
L236:Delete ‘The physical properties of the substrate can affect the transmission of vibrational signals’, which has been mentioned before.
L242:‘which can be transmitted via the plant.’ I really don’t understand that statement.
L246:‘MCaS playback on each leaf stimulated female responses’. This is not mentioned in the results.
L264-265:I really don’t understand that statement.Please state clearly.
Materials and methods
L301:Why did you choose leafhoppers at peak period? Will off-peak leafhoppers matter?
L302:Indicate the plant cultivar of tea plant.
L365:Is the signal collection conducted on the seventh leaf below the bud? please clarify.
L400-406: This should be stated in section 4.3.
L423-424:Were independent variables with weak linear relationships eliminated?In multilinear analysis, variables with multicollinearity need to be selected and eliminated. I suspect there is a misrepresentation here. please clarify.
Conclusion
L445:Replace ‘a’ with ‘this’.
L444-447:‘The attenuation of signal intensity … of male signals by females.’ The premise of this conclusion is that if the signal intensity is below the female's perceptual threshold, the female will not respond to the male. Please modify.
Comments on the Quality of English LanguageSee above the comments
Author Response
This paper is innovative. I think this manuscript should be published in the journal after revision.
Response: Thanks for your valuable and constructive suggestions for our research. We hope that the revisions in the manuscript and our responses will be sufficient to improve our manuscript for publication.
Abstract:
- L14:Delete ‘(e.g., Empoasca onukii)’.
Response: This has been revised as you suggestion. (L 14)
- L15:Supplement the host after Empoasca onukii.
Response: This has been revised as you suggestion. (L 16)
Introduction:
- L42:I think this should be ‘time pattern’her.
Response: Thanks, this has been revised as you suggestion. (L 49)
Results
- L77:Indicate the position of the leaf that female inhabited.
Response: This has been revised as you suggestion. (L 91)
- Figure 1. Behavioral monitoring is not involved in Figure 1A, and supplement the position of the leaf that female inhabited.
Response: We apologize for this bad mistake, which has been corrected.
- L102:‘and was independent of MCaS parameters.’ I think this sentence should be changed to 'intensity of MCaS and duration of MCaS-S2', because the frequency does not change significantly after transmission.
Response: Thanks, this has been revised as you suggestion. (L 118-119)
- L105:I didn't find Table 1 in the manuscript. Why is Table 1 in the supplementary materials? This has a serious impact on the revision of the manuscript.
Response: Sorry for the confusion caused by this low-level error, we have added Table 1 in the manuscript after revision. (L 134)
- L108-110:Conjectural conclusions need to be formulated with care, after all, the function of these sections has not yet been verified.
Response: Thanks, we have changed this to a speculative statement. (L 126)
- L131:Here should be the second or third leaf under the bud.
Response: Thanks. The traits of the second leaf and the third leaf were significantly different, so it doesn’t need to be revised here.
- L134:Replace ‘signal’ with ‘MCaS’.
Response: This has been revised as you suggestion. (L 161)
- L137:Replace ‘signal’ with ‘MCaS’.
Response: This has been revised as you suggestion. (L 164)
- L151-152:Are males always the ones initiating courtship communication?
Response: Yes, the male is the initiator of courtship communication in most cases.
- L171:The case of the acronym 'delay' should be unified (capital and lower-case letter), as should similar issues with other acronyms.
Response: Thank you. We have corrected this throughout the manuscript.
- L192:Replace ‘PTRMCoS-S2’ with ‘PRTMCoS-S2’.
Response: Thank you. We have corrected this error throughout the manuscript.
- L199-200:Simplify this sentence ‘The signal parameters… by female signals’.
Response: This sentence has been revised as ‘The signal parameters of females and males affected each other’. (L 226-227)
- L201:Delete ‘DelayLO indicated the response time of a female to an MCoS’.
Response: This sentence was important so we didn't delete it.
Discussion
- L223-224: I don't understand this sentence, the causal relationship seems to be wrong.
Response: This sentence has been revised as ‘Clarifying host–pest interactions during the courtship communication of E. onukii, an obligate pest of tea plants, has important implications for understanding the mechanisms of adaptation of E. onukii to its host’. (L 251-253)
- L227:Replace ’signal excitation’ with ‘signal excitation of males’.
Response: This sentence was deleted after revision. (L 256)
- L236:Delete ‘The physical properties of the substrate can affect the transmission of vibrational signals’, which has been mentioned before.
Response: This sentence was deleted as your suggestion. (L 269-270)
- L242:‘which can be transmitted via the plant.’ I really don’t understand that statement.
Response: This sentence was revised as ‘which is conducive to transmission via the plant’. (L 275)
- L246:‘MCaS playback on each leaf stimulated female responses.’ This is not mentioned in the results.
Response: Thanks, this statement has been added in the 2.1.2. (L 120-121)
- L264-265:I really don’t understand that statement. Please state clearly.
Response: This sentence was revised as ‘The increase in leaf hardness might lead to more information being carried by MCaSs.’ (L 297-298)
Materials and methods
- L301:Why did you choose leafhoppers at peak period? Will off-peak leafhoppers matter?
Response: Leafhoppers that are not at the peak of courtship rarely emit signals, which affects the implementation of the experiment.
- L302:Indicate the plant cultivar of tea plant.
Response: This has been revised as you suggestion. (L 336)
- L365:Is the signal collection conducted on the seventh leaf below the bud? please clarify.
Response: Yes, the vibrational signals on the leaf that the female inhabited were recorded. This has been clarified after revision. (L 404-406)
- L400-406:This should be stated in section 4.3.
Response: Signal analysis is also data analysis, which does not affect the reader's understanding of the manuscript.
- L423-424:Were independent variables with weak linear relationships eliminated?In multilinear analysis, variables with multicollinearity need to be selected and eliminated. I suspect there is a misrepresentation here. please clarify.
Response: We apologize for this error here. The word ‘weak’ was replaced with ‘high’. (L 470)
Conclusion
- L445:Replace ‘a’ with ‘this’.
Response: This sentence was deleted after revision. (L 495)
- L444-447:‘The attenuation of signal intensity … of male signals by females.’ The premise of this conclusion is that if the signal intensity is below the female's perceptual threshold, the female will not respond to the male. Please modify.
Response: This sentence was deleted after revision. (L 496-497)

Reviewer 3 Report
Comments and Suggestions for Authors
This well-organized manuscript presents an important study that clarifies the impact of leaf traits and age on the variability of mating signals and behaviors in Empoasca onukii
Minor comment:
Line 233. Instead of “on its host“ - “to its host“
Author Response
Comments and Suggestions for Authors
This well-organized manuscript presents an important study that clarifies the impact of leaf traits and age on the variability of mating signals and behaviors in Empoasca onukii.
Response: Thank you for affirming the content of this study.
Minor comment:
Line 233. Instead of “on its host” - “to its host”.
Response: This has been revised as your suggestion. (L253)
